# Echinochrome A Treatment Alleviates Atopic Dermatitis-like Skin Lesions in NC/Nga Mice via IL-4 and IL-13 Suppression

**DOI:** 10.3390/md19110622

**Published:** 2021-11-01

**Authors:** Hyeong Rok Yun, Sang Woo Ahn, Bomin Seol, Elena A. Vasileva, Natalia P. Mishchenko, Sergey A. Fedoreyev, Valentin A. Stonik, Jin Han, Kyung Soo Ko, Byoung Doo Rhee, Jung Eun Seol, Hyoung Kyu Kim

**Affiliations:** 1Department of Physiology, College of Medicine, Cardiovascular and Metabolic Disease Center, Smart Marine Therapeutic Center, Inje University, Busan 47392, Korea; foryou018@naver.com (H.R.Y.); illillil@naver.com (B.S.); phyhanj@inje.ac.kr (J.H.); kskomd@paik.ac.kr (K.S.K.); bdrhee@hanmail.net (B.D.R.); 2Department of Health Sciences and Technology, Graduate School, Inje University, Busan 47392, Korea; 3Department of Dermatology, Inje University Busan Paik Hospital, College of Medicine, Inje University, Busan 47392, Korea; derma09@hanmail.net; 4G.B. Elyakov Pacific Institute of Bioorganic Chemistry, Far-Eastern Branch of the Russian Academy of Science, 690022 Vladivostok, Russia; vasilieva_el_an@mail.ru (E.A.V.); mischenkonp@mail.ru (N.P.M.); fedoreev-s@mail.ru (S.A.F.); stonik@piboc.dvo.ru (V.A.S.)

**Keywords:** atopic dermatitis, echinochrome A, NC/Nga mice, proinflammation, mast cell infiltration

## Abstract

Atopic dermatitis (AD) is a chronic inflammatory skin disease in which skin barrier dysfunction leads to dryness, pruritus, and erythematous lesions. AD is triggered by immune imbalance and oxidative stress. Echinochrome A (Ech A), a natural pigment isolated from sea urchins, exerts antioxidant and beneficial effects in various inflammatory disease models. In the present study, we tested whether Ech A treatment alleviated AD-like skin lesions. We examined the anti-inflammatory effect of Ech A on 2,4-dinitrochlorobenzene (DNCB)-induced AD-like lesions in an NC/Nga mouse model. AD-like skin symptoms were induced by treatment with 1% DNCB for 1 week and 0.4% DNCB for 5 weeks in NC/Nga mice. The results showed that Ech A alleviated AD clinical symptoms, such as edema, erythema, and dryness. Treatment with Ech A induced the recovery of epidermis skin lesions as observed histologically. Tewameter^®^ and Corneometer^®^ measurements indicated that Ech A treatment reduced transepidermal water loss and improved stratum corneum hydration, respectively. Ech A treatment also inhibited inflammatory-response-induced mast cell infiltration in AD-like skin lesions and suppressed the expression of proinflammatory cytokines, such as interferon-γ, interleukin-4, and interleukin-13. Collectively, these results suggest that Ech A may be beneficial for treating AD owing to its anti-inflammatory effects.

## 1. Introduction

Atopic dermatitis (AD) is a skin disease characterized by chronic inflammation of the skin that leads to pruritus, erythematous lesions, skin barrier dysfunction, increased transepidermal water loss (TEWL), and immune-redox disturbances [1,2,3,4]. The pathogenesis of AD remains unclear, but appears to be related to both genetic factors, such as a dysfunctional immune system where type 2 T helper cells secrete an excessive amount of interleukin-4 (IL-4) and interleukin-13 (IL-13), and environmental factors, such as house dust mites [5,6,7,8,9]. In addition, AD lesions are commonly characterized by excessive infiltration of granulated mast cells and an increased leukocyte count. Notably, mast cells are critical for the development of AD-induced inflammatory skin diseases [10]. Mast cells are activated by antigen-specific immunoglobulin E (IgE) through high-affinity IgE receptors (FcεRI). These cells are then recruited into AD skin lesions, where they promote skin hypersensitivity reactions by releasing histamine [11,12,13].

Echinochrome A (Ech A) is a marine-derived polyhydroxynaphthoquinone (Figure 1A) approved for medical use in Russia (PubChem CID: 135457951, C_12_H_10_O_7_) [14,15,16,17,18]. Ech A is known for its anti-inflammatory, antimicrobial, and antioxidative effects [19,20,21,22,23]. In previous studies, Ech A was shown to suppress inflammatory effects, such as those of IL-1b, IL-6, INF-a, and IL-8 in cardiovascular disease in vivo [20]. In addition, treatment with Ech A promotes antioxidative effects, by upregulating the expression of antioxidants, such as glutathione-S-transferase, superoxide dismutase, and catalase in diabetic mice. Furthermore, Ech A treatment facilitates immune system balance by regulating regulatory T cells in an inflammatory bowel disease model [24]. 

In general, AD therapeutic strategies include the use of topical corticosteroids, calcineurin inhibitors, and T-cell inhibitors. However, long-term treatment is known to cause side effects, such as immunosuppression and epidermal barrier dysfunction [25,26,27,28]. Therefore, there is a need to develop effective therapies for AD. Several clinical investigations have shown that natural immune regulators from sea urchins or derivatives may minimize toxicity [29,30,31,32,33,34]. In particular, recent studies have shown that treatment with Ech A facilitates the recovery of damaged dermis in bleomycin-induced scleroderma, including improvement in dermal thickness, collagen density, and hydroxyproline levels [35]. Although Ech A exerts putative anti-inflammatory effects, its role in skin barrier protection in AD-like skin lesions remains unclear.

In this study, we aimed to investigate the therapeutic effect of Ech A on 2,4-dinitrochlorobenzene (DNCB)-treated NC/Nga mice, a known model of AD-like skin lesions.

## 2. Results

### 2.1. Ech A Alleviated DNCB-Induced AD-like Skin Lesions in NC/Nga Mice

To induce AD-like cutaneous skin lesions in NC/Nga mice, we treated them with topical application of cutaneous 1% DNCB three times a week to achieve sensitization. One hundred microliters of 1% DNCB diluted in olive oil and acetone (1:3) was applied. After DNCB sensitization, we used topical application of 0.4% DNCB three times a week for 5 weeks and divided the mice into six groups, with six animals per group: group 1, nontreated (N); group 2, topical application of DNCB treatment (D); group 3, DNCB + topical application of phosphate-buffered saline (PBS) (PT); group 4, DNCB + PBS intraperitoneal (IP) injection (PI); group 5, DNCB + topical application of 0.02% (0.1 mg/kg/200 µL) Ech A (ET); and group 6, DNCB + 0.02% Ech A (0.1 mg/kg/200 µL) IP injection (EI) (Figure 1B). At 4 weeks, the D, PT, and PI groups’ dorsal skin showed mild edema, excoriation, dryness, and erythema (Figure 1C). At 6 weeks, it showed prominent edema, excoriation, dryness, and erythema. However, AD conditions were significantly improved in the ET and EI groups compared with the PT and PI groups at 4 and 6 weeks (Figure 1C).

**Figure 1 marinedrugs-19-00622-f001:**
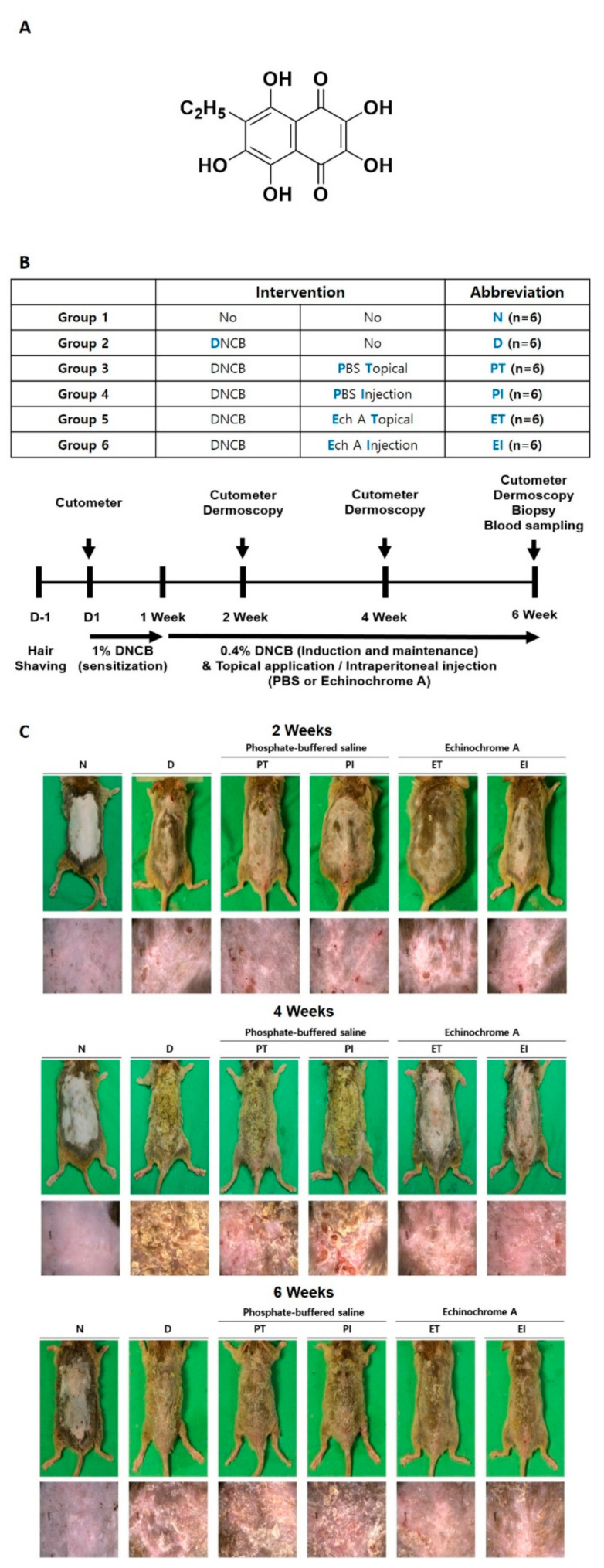
Changes in 2,4-dinitrochlorobenzene (DNCB)-induced atopic dermatitis (AD)-like symptoms in the dorsal skin lesions of the mouse model. (**A**) Structure of echinochrome A (Ech A); (**B**) overall schematic diagram and timeline of experiments. NC/Nga mice were divided into six groups: nontreated (N), DNCB (D), DNCB + PBS topical application (PT), DNCB + PBS intraperitoneal (IP) injection (PI), DNCB + (Ech A) topical application (ET), and DNCB + Ech A IP injection (EI). (**C**) AD-like dorsal skin lesions were observed by dermoscopy every 2 weeks.

### 2.2. Ech A Attenuated Histopathological Conditions of DNCB-Induced Skin Lesion in NC/Nga Mice

To assess the therapeutic effect of Ech A on DNCB-induced AD-like skin tissue thickness in NC/Nga mice, we performed hematoxylin and eosin (H&E) staining. H&E staining showed thickening of the epidermis in the D, PT, and PI groups compared with the N group (Figure 2A,B). In addition, the D, PT, and PI groups showed parakeratotic hyperkeratosis, epidermal hyperplasia, lymphocyte exocytosis, and spongiosis. In contrast, the ET and EI groups exhibited greater recovery of epidermal skin lesions, reduced hyperkeratosis, epidermal hyperplasia, lymphocyte exocytosis, and spongiosis compared with the PT and PI groups (Figure 2A). These results showed that Ech A treatment markedly improved the conditions of the damaged epidermis in the DNCB-induced AD-like NC/Nga mice.

### 2.3. Ech A Improved TEWL and SCH in DNCB-Induced Skin Lesions

Previous studies have demonstrated that TEWL and stratum corneum hydration (SCH) scores are clinically useful for the assessment of AD [36,37,38,39]. Thus, through Tewameter^®^ and Corneometer^®^ measurements, we evaluated whether Ech A treatment on DNCB-induced dorsal skin in NC/Nga mice improved TEWL and SCH. After 2 weeks, TEWL and SCH scores were significantly decreased in the 0.4% DNCB group compared with the N group (Figure 3A,B). However, such scores improved in the ET and EI groups at 4 and 6 weeks compared with the PT and PI groups (Figure 3A,B). Our findings proved that Ech A reduced the loss of moisture and maintained the skin barrier function.

### 2.4. Ech A Reduces Mast Cell Infiltration in AD-like NC/Nga Mice

To investigate the effects on the mast cell infiltration of Ech A in a DNCB-induced AD-like NC/Nga mouse model, we stained the skin tissue samples with toluidine blue. Mast cell infiltration in DNCB-induced skin lesions was significantly increased in the D, PT, and PI groups compared with the N group (Figure 4A,B). In contrast, the ET and EI groups presented a significant reduction in mast cell infiltration compared with the PT and PI groups (Figure 4A,B). These results show that treatment with Ech A reduced mast cell infiltration in DNCB-induced AD-like skin lesions in NC/Nga mice.

### 2.5. Ech A Decreased the Proinflammatory Response in DNCB-Treated NC/Nga Mice

To further confirm the anti-inflammatory effect of Ech A on DNCB-induced skin lesions in NC/Nga mice, we used Western blot analysis to examine whether proinflammatory cytokine levels were suppressed by Ech A treatment. The D group showed increased expression levels of interferon-γ (IFN-γ) compared with the N group (Figure 5A,B). The ET and EI groups showed a remarkable suppression of IFN-γ and IL-13 expression compared with the PT and PI groups (Figure 5A,B,D). IL-4 cytokine expression was reduced in the EI group only (Figure 5C). In addition, ELISA measurements showed that the ET and EI groups had lower IgE serum levels compared with the D group (Appendix A). These results indicate that Ech A treatment reduces the proinflammatory response in DNCB-induced AD-like skin disease.

## 3. Discussion

Several studies have reported that defective skin barrier function induces AD by stimulating keratinocytes and evoking proinflammatory responses [40,41]. Skin barrier dysfunction plays a key role in AD disease by allowing the penetration of allergens through the skin and facilitating their presentation to the local immune effector cells [40,42]. In addition, the pathogenesis of AD due to the loss of moisture is favored by skin sensitivity to various allergens [36,37,38,43,44]. Therefore, effective therapies are needed to suppress immune system activation and skin inflammation leading to AD.

Clinical and experimental studies have shown that immune regulators from natural marine extracts may have therapeutic effects on AD [29,30,34,45]. Based on these reports, we investigated whether Ech A extracted from sea urchins could improve AD-like skin lesions in NC/Nga mice through an increase in skin moisture. According to previous reports, Ech A has beneficial effects, including antioxidative and anti-inflammatory responses and an improved immune system [19,20,21].

Our results showed that Ech A exerts therapeutic effects on DNCB-induced AD lesions in NC/Nga mice by improving skin barrier function and exhibiting anti-inflammatory effects. Ech A treatment suppressed the DNCB-induced allergic response and improved the thickness of the epidermis. In addition, TEWL and SCH scores were improved in the DNCB-induced NC/Nga mouse model treated with Ech A extract. We observed that Ech A treatment decreased inflammatory skin responses, including production of IFN-γ, IL-4, and IL-13, in DNCB-induced skin lesions. We also observed that mast cell infiltration in these lesions was reduced by treatment with Ech A. Our findings suggest that Ech A treatment may have therapeutic effects on AD. Recent studies have shown that the levels of TEWL and SCH are associated with the severity of AD symptoms. Consistently, in our study, Ech A treatment significantly regulated the levels of TEWL and SCH, suggesting that Ech A may help maintain epidermal skin barrier function and regulate the severity of AD.

AD is characterized by type 2 immune responses driven by various cytokines, including IL-4, IL-5, IL-9, IL-13, and IL-25 [46,47]. In particular, IL-4 and IL-13 have been implicated in the pathogenesis of AD. In previous studies, RT-PCR, Western blot, and ELISA analyses showed excessive amounts of IL-4 and IL-13 in AD-like skin lesions, at both the mRNA and protein levels [48,49,50]. Regulation of IL-4 and IL-13 in AD-like skin lesions is believed to play an important role in the development of AD. Previous studies on the therapeutic effect of IL-4 and IL-13 inhibition have proven its efficacy [51,52,53,54,55]. Thus, Ech A treatment, especially IP injection, is expected to have beneficial effects in minimizing inflammation in patients with AD. However, the detailed molecular and biochemical mechanisms underlying the therapeutic effects of Ech A remain to be elucidated, and our findings provide evidence that topical application and IP injection of Ech A effectively alleviate DNCB-induced AD-like skin lesions by suppressing inflammatory cytokines. 

## 4. Conclusions

Our results are the first to show the therapeutic potential of Ech A treatment in alleviating DNCB-induced AD-like skin lesions. We demonstrate that Ech A prevents both skin damage and the proinflammatory effect of IL-4, IL-13, and mast cell infiltration. These findings indicate a promising role for the marine drug Ech A in the treatment of chronic inflammatory skin diseases.

## 5. Materials and Methods

### 5.1. Preparation of Echinochrome A

Histochrome^®^ containing 1% echinochrome A (2,3,5,7,8-pentahydroxy-6-ethyl-1,4-naphthoquinone, pharmaceutical, state registration number PN002362/01-2003) was provided by G.B. Elyakov Pacific Institute of Bioorganic Chemistry FEB RAS, Russia. Histochrome^®^ composition is 1% Ech A in a 0.9% isotonic solution (sodium carbonate and sodium chloride, 37.5 mM). The PBS group was administered 200 µL of PBS via topical application and IP injection. The Ech A group was administered 0.1 mg/kg of Ech A in 200 µL of PBS (approximately 0.02% echinochrome A) [56,57,58,59].

### 5.2. DNCB-Induced AD-like Skin Disease Model

Six-week-old male NC/Nga mice were purchased from Orient Bio (Daejeon, Korea). The mice were acclimatized under 22 ± 2 °C and 45 ± 5% humidity with a 12/12 h dark/light rotation for 1 week. The animal study was approved and carried out in accordance with the Institutional Animal Care and Use Committee of Inje University (approval number 2020-002). To induce the development of AD-like epidermal skin lesions, we treated the NC/Nga mice with topical application of 100 µL of 1% DNCB three times a week for 2 weeks, followed by topical application of 100 µL of 0.4% DNCB three times a week for 5 weeks.

### 5.3. Measurement of TEWL and SCH

After sensitization with 1% DNCB and 0.4% DNCB, TEWL and SCH scores were measured using a Tewameter^®^ and a Cutometer^®^ MPA 580 every 2 weeks (Courage + Khazaka Electronic GmbH, Cologne, Germany). The measuring probe was placed on the middle portion of the dorsal skin of the mice, which were anesthetized by inhalation of a low concentration of isoflurane. The levels of TEWL were analyzed at the plateau of the estimation graph, while SCH levels were taken as the mean value of three measurements. 

### 5.4. Histological Analysis

To collect the dorsal skin, we used CO_2_ gas to euthanize all the mice. Portions of the dorsal skin biopsies were fixed in 4% paraformaldehyde and then moved to a 70% ethyl alcohol solution at 20–22 °C for 24 h. The samples were embedded in paraffin wax and sectioned into 4 µm slices. The skin sections were stained with hematoxylin and eosin (H&E) and toluidine blue. Stained slides were observed using a NanoZoomer Digital Pathology slide scanner and HCImage software (Hamamatsu Photonics, Milan, Italy).

### 5.5. Observation of AD-like Skin Condition

To check the severity of the conditions of AD-like dorsal skin, photographs were taken with a dermoscopy camera (DermLite™ DLCAM, DermLite, San Juan Capistrano, CA, USA) every 2 weeks. The mice were photographed under anesthesia, which was administered through brief inhalation of isoflurane.

### 5.6. Western Blot

Epidermal dorsal skin samples were lysed with radioimmunoprecipitation assay buffer (50 mM Tris-HCl, pH 7.5, 150 mM NaCl, 0.5% sodium deoxycholate, 1% Triton X-100, 2 mM EDTA, and 0.1% SDS) supplemented with protease and phosphatase inhibitors (200 mM PMSF). Protein concentration was determined using a Bio-Rad DC protein assay (Bio-Rad Laboratories, Inc., Hercules, CA, USA). Lysates were separated by 8–15% sodium dodecyl sulfate polyacrylamide gel electrophoresis (SDS-PAGE) and transferred onto a nitrocellulose membrane (GE Healthcare, Chicago, IL, USA) for 100 min at 110 V and 4 °C. Membranes were blocked with 5% skim milk in TBST (20 mmol/L Tris-HCl, pH 7.5, 50 mmol/L NaCl, and 0.1% Tween 20). After blocking, membranes were incubated overnight with primary antibodies diluted to 1:1000–1:5000 in 3% bovine serum albumin (BSA) in TBST. After washing three times for 15 min, the membranes were incubated for 1 h with horseradish peroxidase-labeled secondary antibody diluted to 1:1000–1:5000 in 3% BSA [60]. The protein bands were visualized using an enhanced chemiluminescence kit (Santa Cruz Biotechnology, Dallas, TX, USA) and observed with an AI 600 Imager (GE Healthcare, Chicago, IL, United States). 

### 5.7. Statistical Analysis

All data are presented as the mean ± SEM of at least three independent experiments. Statistical comparisons were analyzed using one-way analysis of variance (ANOVA), followed by Tukey’s post hoc test for multiple comparisons (GraphPad Prism 8.0, GraphPad Software Inc., San Diego, CA, USA). Statistical significance was set at *p* < 0.05.

## Figures and Tables

**Figure 2 marinedrugs-19-00622-f002:**
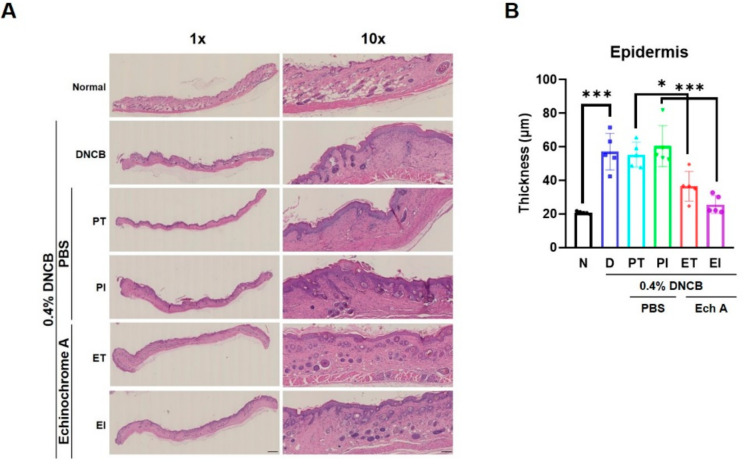
Histopathological images showing how Ech A improves the condition of damaged epidermis in DNCB-induced AD-like lesions in NC/Nga mice. (**A**) Skin sections were stained with hematoxylin and eosin (H&E) for the measurement of epidermal thickness in the DNCB-induced AD mouse model; (**B**) the thickness of the epidermis was measured with a NanoZoomer. The data shown in the graphs represent the mean ± SEM. *** *p* < 0.001, * *p* < 0.05. Scale bar = 100 μm.

**Figure 3 marinedrugs-19-00622-f003:**
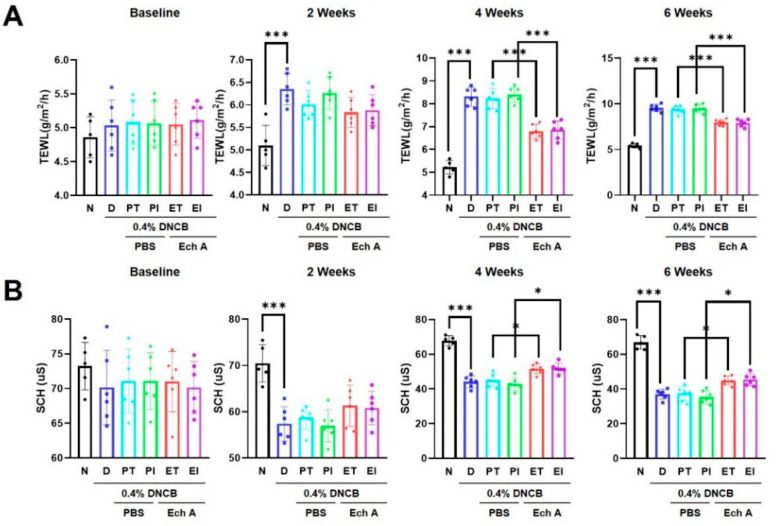
Effect of Ech A on transepidermal water loss (TEWL) and stratum corneum hydration (SCH) scores in DNCB-induced AD-like NC/Nga mice. (**A**) Levels of TEWL in the skin epidermis were evaluated every 2 weeks through Tewameter^®^ measurements; (**B**) SCH score was determined through Corneometer^®^ measurements every 2 weeks in the DNCB-induced skin lesions. The data shown in the graphs represent the mean ± SEM (*n* = 6). *** *p* < 0.001, * *p* < 0.05.

**Figure 4 marinedrugs-19-00622-f004:**
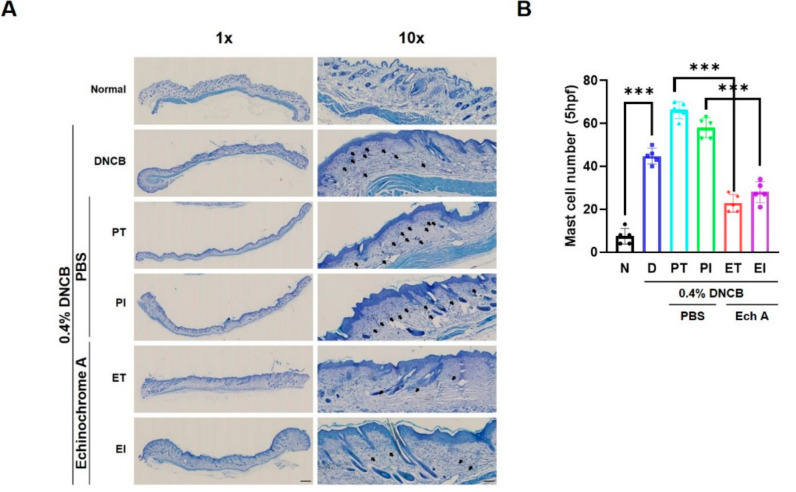
Ech A reduces mast cell infiltration in AD-like NC/Nga mice. (**A**) Histological observation of toluidine-blue-stained samples showed reduced infiltration of mast cells in the ET and EI groups compared with the DNCB-treated groups; (**B**) the number of infiltrated mast cells was counted in five representative high-power fields by means of a NanoZoomer (*n* = 6). The data shown in the graphs represent the mean ± SEM. *** *p* < 0.001. Scale bar = 100 μm.

**Figure 5 marinedrugs-19-00622-f005:**
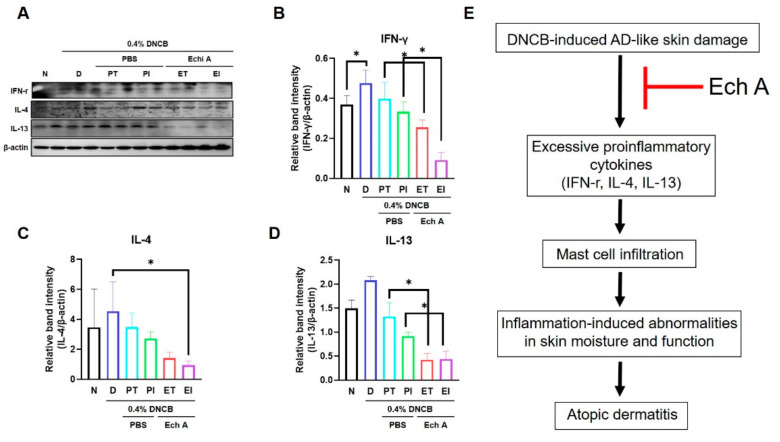
Ech A treatment showed an anti-inflammatory effect in the DNCB-induced AD-like mouse model. (**A**) Immunoblot analysis of protein expression related to proinflammation; (**B**–**D**) relative band intensity of skin interferon-γ (IFN-γ), interleukin-4 (IL-4), and interleukin-13 (IL-13) in AD-like skin lesions assessed through analysis with ImageJ. β-actin served as an internal standard; (**E**) scheme of the suppression mechanism of Ech A treatment in AD-like skin lesions. The data shown in the graphs represent the mean ± SEM of three independent experiments. * *p* < 0.05.

## Data Availability

This article does not data and the data availability policy is not applicable to the article.

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
