# Peer review of "Echinochrome A Treatment Alleviates Atopic Dermatitis-like Skin Lesions in NC/Nga Mice via IL-4 and IL-13 Suppression"

_marinedrugs, 2021, doi:10.3390/md19110622_

Round 1

Reviewer 1 Report

The manuscript by Hyeong Rok Yun is dedicated to the efficacy of treatment of atopic dermatitis-like skin lesions in mice. I think that the manuscript could be suitable for Marine drugs after revisions according to the next suggestions:

  1. Line 0-51 - I suggest supporting with a more recent reference (https://doi.org/10.1007/s11101-018-9547-3).
  2. It seems that Echinochrome A is only a marine-derived polyhydroxynaphthaquinone approved for medicinal use (https://doi.org/10.1016/j.jep.2019.111933)
  3. In lines 78-79: why PBS) was selected as a control?
  4. In lines 79-80: please indicate doses of Ech A for topical application and injection. Please justify doses selection.
  5. Fig. 5E: The scheme is not clear. It seems from the scheme, that Ech A application led to Atopic dermatitis (which is wrong, I suppose). Please revise the scheme.
  6. In lines 195-197: did the authors used 2,3,5,7,8-pentahydroxy-6-ethyl-1,4-naphthoquinone or salt? Which concentrations of Ech A were used in experiments on animals after dissolution in PBS?
  7. I suggest providing the conclusion in a separate section. Please support the conclusion with real data in numbers.

Reviewer 2 Report

The manuscript - Echinochrome A treatment alleviates Atopic Dermatitis-like skin lesion in NC/Nga mice by suppression of IL-4 and IL-13 – studied the effect of Echinochrome A on skin lesion. The aim of this study is relative clear and solid.  But there are some limitations about this manuscript. After the authors solve them perfectly, I suggest this manuscript could be published. My detailed comments are as follows:

  1. In fig 3, a blank space should be added between 4 and weeks, and the words below the x-coordinate should be presented in bigger font.
  2. The description about the structural information of Echinochrome A should be added.
  3. Line 126, what is N Group, please clear it.
  4. Particular, in fig 5, the cytokines were measured by WB, it is not exact, and the ELISA measurement of the cytokines should be performed.

Round 2

Reviewer 1 Report

Authors have responded to my questions and have revised the paper. The manuscript could be accepted in the present form.

Author Response

Thank you for the valuable and constructive  comment.  

Reviewer 2 Report

Moderate English changes were required.

Author Response

Dear Reviewer

Thank you for the valuable comment. As follow your comment, we did English editing through Editage service. I attached the certificated of English editing
